# Local and global crosstalk among heterochromatin marks drives DNA methylome patterning in Arabidopsis

Taiko Kim To [1,2✉], Chikae Yamasaki[1], Shoko Oda[1], Sayaka Tominaga[1], Akie Kobayashi[2], Yoshiaki Tarutani[2] & Tetsuji Kakutani [1,2✉]

Transposable elements (TEs) are robustly silenced by multiple epigenetic marks, but dynamics of crosstalk among these marks remains enigmatic. In Arabidopsis, TEs are silenced by cytosine methylation in both CpG and non-CpG contexts (mCG and mCH) and histone H3 lysine 9 methylation (H3K9me). While mCH and H3K9me are mutually dependent for their maintenance, mCG and mCH/H3K9me are independently maintained. Here, we show that establishment, rather than maintenance, of mCH depends on mCG, accounting for the synergistic colocalization of these silent marks in TEs. When mCG is lost, establishment of mCH is abolished in TEs. mCG also guides mCH in active genes, though the resulting mCH/H3K9me is removed thereafter. Unexpectedly, targeting efficiency of mCH depends on relative, rather than absolute, levels of mCG within the genome, suggesting underlying global negative controls. We propose that local positive feedback in heterochromatin dynamics, together with global negative feedback, drive robust and balanced DNA methylome patterning.

[1] Department of Biological Sciences, The University of Tokyo, Tokyo 113-0033, Japan. [2] Department of Integrated Genetics, National Institute of Genetics, Mishima, Shizuoka 411-8540, Japan. ✉email: tkt@bs.s.u-tokyo.ac.jp; tkak@bs.s.u-tokyo.ac.jp

Large genomes of vertebrates and plants contain substantial amounts of transposable elements (TEs) and their derivatives. As TEs are a potential threat to genome stability and proper gene expression, they are silenced by epigenetic mechanisms such as cytosine methylation and histone H3 methylation at lysine 9 (H3K9me)[1–4]. In plant genomes, methylated cytosines are enriched in TEs for both CpG and non-CpG (or CpH, where H can be A, T, or C) contexts[5,6]. In Arabidopsis, methylation at CpG sites (mCG) is maintained by a DNA methyltransferase (MTase) called MET1 (METHYLTRANSFERASE 1)[7,8]. Methylation at CpH sites (mCH) is catalyzed by another class of DNA MTases, CHROMO-METHYLASE 2 and 3 (CMT2 and CMT3)[9–12]. Within mCH, symmetric mCHG and asymmetric mCHH are often analyzed separately, because their controls are different; mCHG is catalyzed by CMT3 and CMT2, while mCHH is catalyzed by CMT2[11,12]. These CMTs are recruited to regions with H3K9me. H3K9me is catalyzed by SUVH4, SUVH5, and SUVH6, and these H3K9 MTases, in turn, are recruited to regions with mCH[10,13,14], generating a self-reinforcing positive feedback loop[15]. By this positive feedback, mCH and H3K9me are maintained through cell divisions.

While H3K9me and mCH depend on each other, their relationship to mCG has received less attention. Mutations in the CpG MTase gene *MET1* have only minor effects on mCH and H3K9me; similarly, mutations in CpH MTases CMTs or H3K9 MTases SUVHs also have only minor effects on mCG[5,6,13,16–18]. Thus, these two layers of modifications are maintained largely independently.

Although the two layers of modifications, mCG, and mCH/H3K9me, are maintained independently, they are associated with each other during de novo establishment. This association in the establishment can be seen in both RNAi-dependent and -independent pathways. Plants can methylate both CpG and CpH sites in an RNAi-based pathway, called RdDM (RNA-directed DNA methylation). RdDM is a mechanism to trigger mCG and mCH by de novo DNA MTase DRM2, and its targeting depends on siRNAs and siRNA-associating RNAi components[19,20]. In addition to this well-investigated RNAi-based pathway, we have recently identified a very robust and precise RNAi-independent pathway to establish mCH and H3K9me de novo in coding regions of TEs (TE genes); both H3K9me and mCH are lost simultaneously in mutants for SUVHs or CMTs, but these marks recover efficiently and precisely in coding regions of TEs (TE genes) after reintroduction of the wild-type alleles[18]. Unexpectedly, this recovery of mCH/H3K9me is independent of the RNAi-based de novo DNA methylation machinery, such as DRM2 or RNA-dependent RNA polymerases. However, the recovery is inefficient in TE genes that have lost mCG. These results suggest that mCG might induce de novo establishment of mCH and H3K9me in the RNAi-independent pathway, although the causative link between them remains to be tested.

Although both mCG and mCH/H3K9me are enriched in TEs, their distribution patterns differ in active genes. In addition to TEs, about 20% of active genes have mCG in their internal regions (gene bodies)[21–24]. In contrast, mCH and H3K9me are found almost exclusively in TEs[5,6]. A factor contributing to the exclusion of mCH/H3K9me from active genes is a Jumonji domain-containing histone demethylase gene, *INCREASE IN BONSAI METHYLATION 1* (*IBM1*)[25–27]; in *ibm1* mutants, H3K9me and mCH accumulate in expressed genes. Interestingly, the genic H3K9me and mCH in the *ibm1* mutant background are found in genes with body mCG[27]. Thus, mCG might direct de novo mCH in genes, as well as in TEs. However, a causative link between mCG and mCH/H3K9me remains to be examined in genes as well as in TEs.

Here, we directly examine whether mCG is necessary for the establishment of mCH by using a mutation of CpG MTase gene *MET1*. The results reveal that TE genes that lose mCG by the *met1* mutation fail to establish mCH. In addition to the effect on TE genes, the *met1*-induced loss of mCG compromises genic mCH accumulating in the background of *ibm1* mutant. Unexpectedly, the targeting efficiency of genic mCH depends on relative level of mCG within the genome, rather than the absolute levels of mCG. This effect is seen in the mCG-directed mCH for both active genes and TE genes. In addition, mCH in TE genes and the *ibm*1-induced genic mCH affect each other negatively, suggesting global negative feedback in the heterochromatin dynamics. Based on these and previous results, we propose that global negative feedback and local positive feedback of heterochromatin marks, combined with H3K9 demethylation in active genes, results in robust and balanced differentiation of silent and active genomic regions.

## Results

**Loss of MET1 function abolishes establishment of mCH in TE genes.** Because mCH and H3K9me depend on each other, both modifications are lost both in the *cmt2 cmt3* double mutant of the mCH MTase genes (hereafter referred to as *cc*) and in the *suvh4 suvh5 suvh6* triple mutant of the H3K9 MTase genes (hereafter referred to as *sss*). The F1 progeny between these mutants inherited genomes without H3K9me and mCH, but all these mutated genes (*CMT*s and *SUVH*s) are complemented by functional wild-type alleles in heterozygous states in the F1 (Fig. 1a). We previously showed that, in the F1 progenies, mCH and H3K9me recover efficiently despite their absence in the parents (Fig. 1a, c, i)[18]. This efficient recovery is independent of RNAi but inefficient in TE genes that lack mCG at the transcription start site.

To directly investigate the possible contribution of mCG to the establishment of mCH, we conducted the same genetic experiments in the mutant background of CpG methyltransferase gene *MET1* (Fig. 1b). We used the incomplete loss-of-function allele, *met1-1*[28], because the null allele of *met1* mutants in combination with mCH mutants results in severe developmental defects and infertility[29]. We generated the *met1-1 cmt2 cmt3* and the *met1-1 suvh4 suvh5 suvh6* mutants (hereafter referred to as *mcc* and *msss*, respectively) and performed a genetic cross between them to obtain the F1 plants (hereafter we call them as *m*F1; Fig. 1b), which can be compared to the original F1 plants between *cc* and *sss* (Fig. 1a). In contrast to the efficient recovery of mCH in F1 in the wild-type *MET1* background (Fig. 1c, d, i, j), the *m*F1 plants showed severe defects in the restoration of mCH in TE genes for both CHG (Fig. 1e–h) and CHH (Fig. 1k–n) contexts.

**Loss of mCG is associated with loss of mCH recovery.** Next, we examined if the failure of mCH recovery in the *m*F1 was associated with the loss of mCG. Because *met1-1* is an incomplete loss-of-function allele, mCG remains in many, although not all, of TE genes[28,30] (Supplementary Fig. 1a). Likewise, some TE genes kept mCG in the *mcc*, *msss,* and the *m*F1 (Supplementary Fig. 1a). Thus, we compared the residual mCG and the efficiency of mCH recovery in the *m*F1. Indeed, mCH recovery was found where mCG remains (Fig. 2a, b; Supplementary Fig. 2a, b; Supplementary Fig. 3a–c). The residual mCG of the TE genes in the *m*F1 is proportionally associated with their mCH recovery (Fig. 2c, d). TE genes with loss of mCG did not recover mCH (Fig. 2d), further demonstrating the contribution of mCG to mCH establishment. In addition, although the simultaneous loss of mCG and mCH in *msss* or *mcc* mutants results in transcriptional derepression in many TE genes, the mCG-dependent mCH recovery in the *m*F1 (Fig. 2d) induces robust re-silencing of them

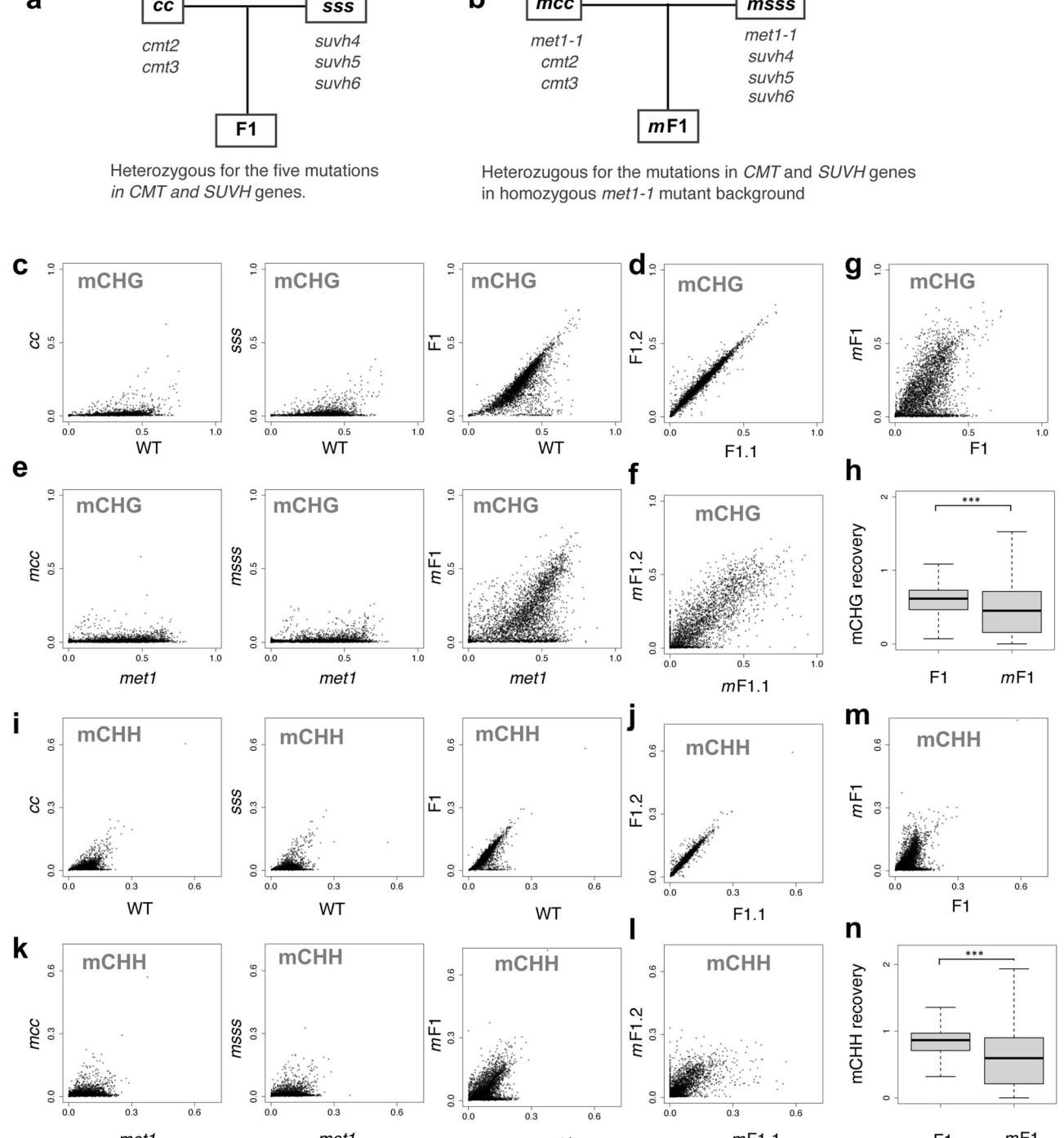

(Fig. 2e), demonstrating the importance of recovered mCH in TE silencing.

**Genic mCH is directed to regions with relatively high levels of mCG.** Next, we asked if mCG in genes also affects their mCH. In wild-type plants, H3K9me2 and mCH are excluded from genes and that depends on the histone demethylase IBM1; in the *ibm1* mutant, H3K9me2, as well as mCH, accumulate in genic regions[25–27]. Importantly, the genic mCH in the *ibm1* mutant correlates with the presence of mCG[26,27]. To see if mCG in gene bodies is responsible for the mCH in the *ibm1* background, we generated the double mutants of *met1-1* and *ibm1-4* (Fig. 3a, bottom left, *met1 ibm1*, hereafter referred to as *mi*) and examined

the effect of the *met1* mutation on *ibm1*-induced genic mCH accumulation. In both *met1-1* and *mi* plants, genic mCG was drastically reduced, but they still had some residual mCG (Fig. 3b, top panel; Supplementary Fig. 1b; Supplementary Fig. 4). Unexpectedly, although the genic mCG is low in the *mi* mutant, a significant increase of genic mCH was detected (Fig. 3b, middle and bottom panels; Supplementary Fig. 4). Nonetheless, the genic mCH in *mi* plants correlated with levels of their residual mCG (Fig. 3c–e), suggesting that the relative, rather than absolute, level of mCG within the genome is critical. In other words, mCH is targeted to regions with mCG, but this targeting seems much enhanced when mCG levels are low in the other regions of the genome. The spectrum of mCH and mCG in *mi* plants differed

**Fig. 1 Loss of MET1 function abolishes establishment of mCH in TE genes. a, b** Scheme of genetic crosses. Mutant of CpH MTases (*cmt2 cmt3*: *cc*) and mutant of H3K9 MTases (*suvh4 suvh5 suvh6*: *sss*) were crossed to generate F1, which are heterozygous for all the mutated genes (**a**). The analogous cross was done between F0 of *met1-1* mutant background (*mcc* and *msss*) to generate F1 with homozygous *met1-1* mutation (*m*F1) (**b**). **c** mCHG recovery in the F1. mCHG level of each TE gene in the mutants, *cc* (left), *sss* (middle), and the F1 (right), compared to a wild-type plant (WT). The results are reproduction of published results (GSE148753[18]). **d** A comparison of mCHG between the two individual F1 plants. **e** mCHG recovery in the *m*F1. mCHG level of each TE gene in the mutants *mcc* (left), *msss* (middle), and the *m*F1 (right) compared to a *met1-1* mutant. **f** A comparison of mCHG between the two individual *m*F1 plants. **g** A comparison of mCHG level between the F1 and *m*F1 plants. **h** Comparison of the efficiency of mCHG recovery in F1 and *m*F1 plants (***$P < 0.0001$, Wilcoxon signed-rank test, two-sided). The efficiency of recovery was calculated as F1/WT and *m*F1/*met1*, respectively. To avoid division by values near zero, TE genes with mCHG ($>0.1$) in both WT and *met1* mutant were used ($n = 3212$). Outliers are not shown. The centerline and box edges represent quartiles and whiskers range 1.5 times of the interquartile from the box edges. **i–m** mCHH recovery of TE genes in the F1 and *m*F1 in the format of (**c**)–(**g**), respectively. **n** Comparison of the efficiency of mCHH recovery in F1 and *m*F1 plants (***$P < 0.0001$, Wilcoxon signed-rank test, two-sided). The efficiency of recovery was calculated as F1/WT and *m*F1/*met1*, respectively. To avoid division by values near zero, TE genes with mCHH ($>0.03$) in both WT and *met1* mutant were used ($n = 2915$). Outliers are not shown. The original data for WT, *cc*, *sss* and F1 are from GSE148753[18]. The centerline and box edges represent quartiles and whiskers range 1.5 times of the interquartile from the box edges. Source data underlying Fig. 1 are provided as a Source Data file.

from that in *ibm1* plants and the spectrum differed even between different *mi* individuals (Supplementary Fig. 5a–c; Supplementary Fig. 6). Importantly, the genic mCH in each of *mi* plants correlated best with mCG level of the same plant but much less to those of the other plants (Fig. 3f; Supplementary Fig. 5d), suggesting that targeting of mCH is controlled by mCG, rather than other intrinsic properties of each gene.

**Association of genic mCG and mCH in epigenetic variant individuals.** mCH accumulated preferentially in genes with relatively high levels of mCG in *mi* as in *ibm1* single mutant plants. Unexpectedly, however, the level of mCG required for the accumulation of mCH seems much lower in *mi* than that in *ibm1*; 0.1 of mCHG was achieved when genes have about 0.04 of mCG level in *mi*, while more than 0.3 of mCG level in *ibm1* single mutant (Fig. 3d). One possible explanation for the difference is that it is attributable to the global reduction of mCG in *met1* background; in other words, the accumulation of genic mCH in *ibm1* mutant background may depend on the relative, rather than absolute, level of mCG. If the relative amount of mCG is really critical for the mCH, local loss of mCG is expected to show a clearer and stronger effect. We next tested this hypothesis using materials with local and heritable loss of mCG, as stated below.

In plants, variations in mCG patterns tends to be transmitted stably over generations[31–34]. That results in heritable loss of genic mCG detected among wild-type *MET1/MET1* siblings in the progeny of *MET1/met1* heterozygotes, and the spectrum of mCG loss is variable among individuals[35,36]. Consistent with these previous results, loss of genic mCG was detected in the *MET1/MET1 ibm1/ibm1* plants (hereafter referred to as *Mi*) originated from *MET1/met1-1 IBM1/ibm1* double-heterozygotes (Fig. 3a, bottom right and Supplementary Fig. 1c), and the spectrum and degree of mCG differed among individual *Mi* plants (Fig. 4a). We used such heritable mCG variation among individuals to examine the association between local mCG loss and the ectopic mCH seen in the *ibm1* mutants.

In the two *Mi* siblings, genes with high mCG in only one of the two individuals (red and blue in Fig. 4a) show specific accumulation of mCH in that individual (Fig. 4b, c; Supplementary Fig. 6b, c). In addition, genes with low mCG in both of the two individuals (green in Fig. 4a) show a consistent lack of mCH accumulation (Fig. 4b; Supplementary Fig. 6c). These results further demonstrate that genic mCH detected in the absence of *IBM1* function depends on mCG.

**mCH is controlled not only locally but also globally.** The results above indicate that the relative, rather than the absolute, mCG level is critical for targeting of genic mCH seen in the *ibm1* mutant background. The mCH levels may be controlled by a global negative feedback mechanism, which would explain the significant genic mCH in *mi* plants despite its low mCG. Analogous effects can also be detected in the mCG-directed mCH of the TE genes in *m*F1. In *m*F1, mCH recovery was more efficient in TE genes with a relatively high level of mCG (Fig. 2c, d); and interestingly, in these TE genes with a relatively high level of mCG, mCH recovery tend to be more efficient in *m*F1 than in F1 (Fig. 2d, middle and right), despite less mCG in *m*F1 than in F1 (Fig. 2d, left). We further examined properties of TE genes with significantly higher mCH in *m*F1 than in F1 (TE genes colored in blue in Supplementary Fig. 3a, b); importantly, the efficient mCH recovery in *m*F1 coincides with the fact that these TE genes have a relatively high level of mCG within the *m*F1 genome, but not within the F1 genome (Supplementary Fig. 3d, "rank" panel). Thus, global negative feedback seems to function for mCG-directed mCH in both active genes and TE genes.

In agreement with this interpretation, we have previously reported results suggesting global negative feedback to control heterochromatin marks, using another Arabidopsis mutant *ddm1* (*decrease in DNA methylation 1*)[11,37,38]. The *ddm1* mutation induces loss of mCG and mCH in heterochromatic TE genes[11]. The *ddm1*-induced loss of mC in TEs is associated with ectopic and stochastic gain of mC in other loci including genic regions[39,40]. Importantly, the descendants from a cross between *ddm1* and wild-type plants in wild-type *DDM1* genotype (i.e. *ddm1*-epiRIL) show the ectopic methylation even in regions originating from wild-type *DDM1* parents. This ectopic methylation correlates with the amount of chromosome regions inherited from the *ddm1* parent, further demonstrating that loss of DNA methylation in TE genes induces genic mCH in *trans*[40]. The ectopic mCH is also induced in a *met1* mutant[32], consistent with the idea that global loss of mCG accelerates targeting of mCH/ H3K9me machinery via a global negative feedback mechanism.

Negative interaction could also be detected between mCH in TE genes and the *ibm1*-induced genic mCH. Gain of genic mCH in *ibm1* mutant induced reduction of mCH in TE genes (Fig. 5a; Supplementary Fig. 7a). In the *ibm1* mutant, the genic mCH accumulate progressively over generations[40,41], and the reduction of mCH in TE genes in *ibm1* was also progressive over generations (Fig. 5a; Supplementary Fig. 7a), consistent with the idea that mCH levels are controlled globally by negative feedback. As the opposite direction of interaction, we also examined if *ddm1*-induced loss of mCH in TE genes affects *ibm1*-induced genic mCH. Indeed, the genic mCH in *ibm1* mutation was much enhanced in the *ddm1 ibm1* double mutant (Fig. 5b, c; Supplementary Fig. 7b, c). Taken together, these results suggest that mCH levels are controlled by global negative feedback.

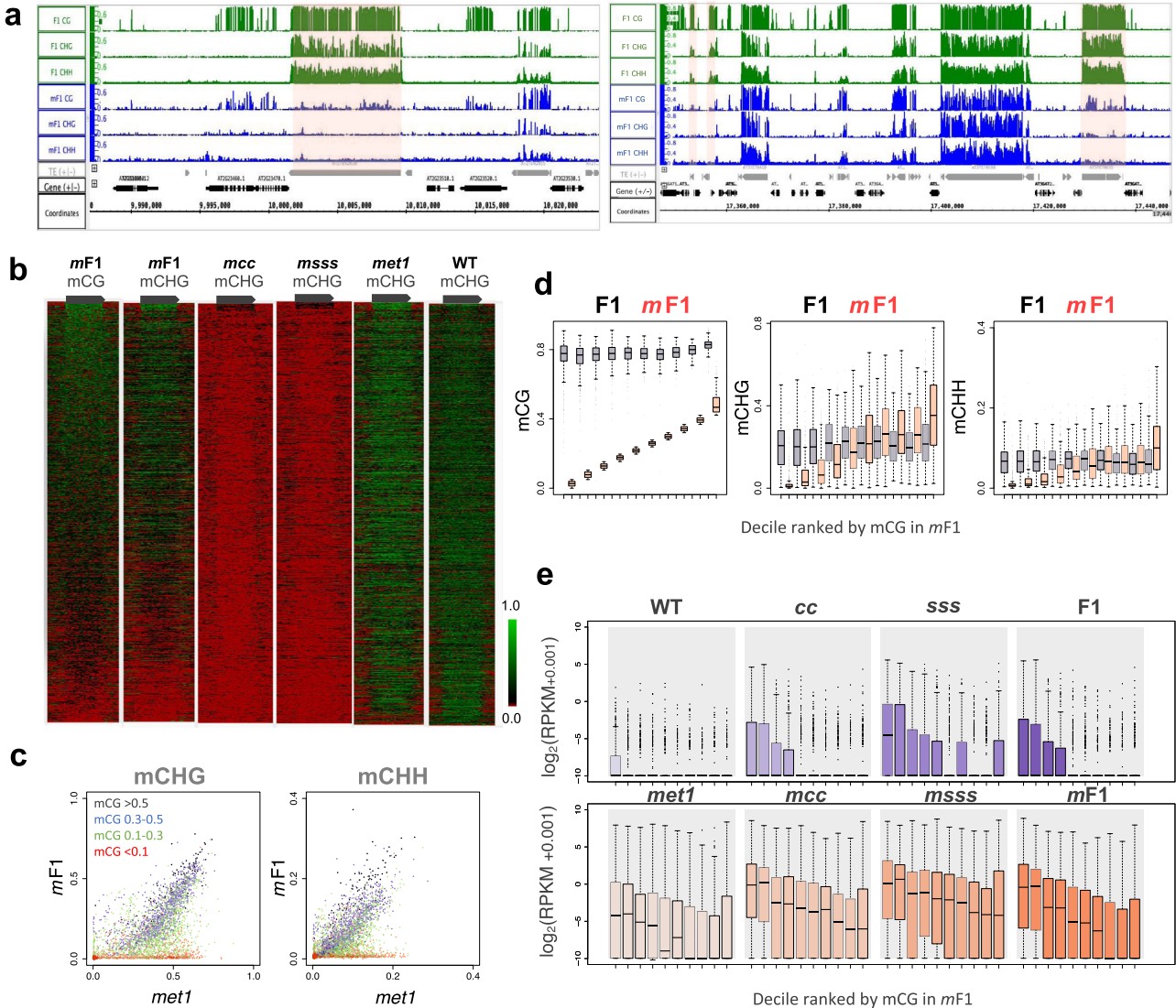

**Fig. 2 Loss of mCG is associated with loss of mCH recovery. a** Genome Browser view of mCG, mCHG, and mCHH in F1 and *m*F1 plants. TEs with loss of mCG fail to restore mCH in *m*F1 (highlighted in pink). The regions of Chr2/9,987,000–10,024,000 (left) and Chr1/16,420,326–16,439,454 (right) are shown. Gray and black arrows represent TEs and genes, respectively. **b** mCHG recovers in TE genes with remaining mCG. Each TE gene (length > 1000; n = 2422) is aligned according to the mCG levels in *m*F1 (left) and the methylation levels of TE genes and their flanking regions (2 kb) are shown in the form of a heatmap for each genotype (see Methods). Gray arrow represents TE gene from TSS to TTS. **c** Recovery of mCHG (left) and mCHH (right) is associated with the residual mCG in *m*F1. The scatter plots shown in the right panel of Fig. 1e and Fig. 1k were colored according to the residual mCG in *m*F1. Red: <0.1, green: 0.1–0.3, blue: 0.3–0.5, and black: >0.5 for mCG in *m*F1. **d** Boxplots showing the association of mCH recovery in *m*F1 with their residual mCG (**d**), but not with their length (Supplementary Fig. 2c). TE genes are divided into deciles according to the mCG levels in *m*F1, and their mCG (left), mCHG (middle), and mCHH (right) levels in F1 and *m*F1 plants are shown. TE genes with mCHG (>0.1) in both WT and *met1-1* were analyzed (n = 3200). The *x*-axis represents mCG levels in *m*F1 divided into deciles and ordered from lower to higher. The centerline and box edges represent quartiles and whiskers range 1.5 times of the interquartile from the box edges. **e** Boxplots showing RNA expression levels of indicated genotypes. TE genes were divided into deciles according to the mCG levels in *m*F1 as shown in Fig. 2d (left), and their RPKM-normalized expression levels (+0.001) are shown in log2. TE genes with mCHG (>0.1) in both WT and *met1-1* were analyzed (n = 3200). Two biological replicates were examined for each genotype and one representative is shown as the two replicates show the same trend. The centerline and box edges represent quartiles and whiskers range 1.5 times of the interquartile from the box edges. Source data underlying Fig. 2c, d are provided as a Source Data file.

## Discussion

In plant genomes, both mCG and H3K9me2/mCH are important for silencing TEs, and both are enriched in TEs. Paradoxically, however, these two layers of modifications are maintained almost independently. Here, we showed that mCG directs the establishment of mCH, even though their maintenance is rather independent. Thus, we propose that colocalization of mCG and mCH in heterochromatin reflects this mechanistic link. Importantly, the very robust targeting of mCH we examined in this study is independent of RNAi for both TE genes[18] and active genes[26].

In contrast, RNAi is involved in the targeting of mCG. The dynamics of mCG have been examined after its loss in *met1* mutation and reintroduction of a functional *MET1* gene[18,30,42]. In those systems, TE genes with efficient mCG recovery are associated not only with mCH but also with siRNAs, suggesting that RNAi and/or mCH also enhance the establishment of mCG. When methylation is lost in both contexts in the *ddm1* mutants,

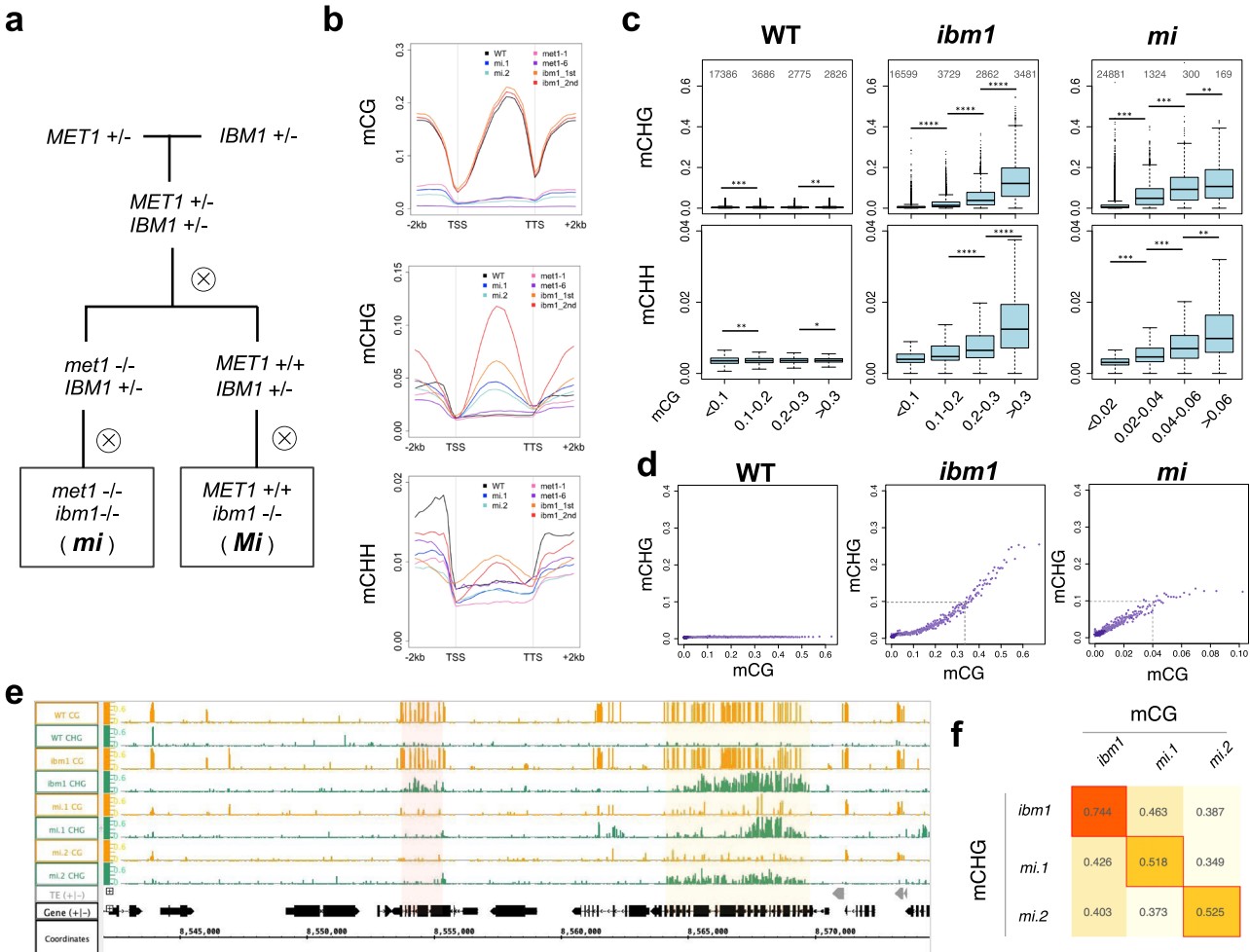

**Fig. 3 Genic mCH is directed to regions with relatively high levels of mCG. a** Materials used. Starting from the cross between plants having *met1-1* (*met1*) or *ibm1-4* (*ibm1*) mutation in the heterozygous state, the double heterozygote was self-pollinated to fix *met1* homozygote and *MET1* wild-type in the F2 generation. The *ibm1* mutation was fixed to be homozygous in the F3 generation. The F3 plants with homozygous *met1-1* mutation (*mi*) and homozygous wild-type *MET1* (*Mi*) were used in this study. **b** Averaged mCG, mCHG, and mCHH profiles over genes in a WT plant, *ibm1*, *met1 ibm1* (*mi*), and *met1* mutants. Values for *met1-6*, a null mutant of *MET1*, are shown for comparison (GSE148753). **c** mCH levels are correlated with the mCG levels in both *ibm1* and *mi*. Genes are divided according to the mCG levels in the indicated genotypes and corresponding mCH levels are shown in boxplots (*P < 0.01, **P < 0.001, ***P < 0.0001, Wilcoxon rank-sum test, two-sided). The number of genes in each class is shown on the top. The centerline and box edges represent quartiles and whiskers range 1.5 times of the interquartile from the box edges. **d** The genes were sorted by the mCG levels of indicated genotypes, grouped in bins with 50 genes, and averaged mCHG levels in the bins are plotted against averaged mCG levels in the corresponding bins. Mean value is plotted for each bin. **e** Browser view of mCG (orange) and mCHG (green) in WT, *ibm1*, and two individual *mi* mutants. The region of Chr1/ 8,542,000–8,574,500 is shown. Gray and black arrows represent TEs and genes, respectively. The yellow shadow indicates an example of genic mCHG commonly seen in *ibm1* and *mi* mutants. It has residual mCG in the *mi* plants. The pink shadow indicates an example of genic mCHG seen in *ibm1* but not in *mi* mutants. **f** Correlation analysis of mCHG and mCG in the *ibm1* and two *mi* individual plants. The best correlated in the individuals are boxed in red. The number in the box represents Pearson's correlation coefficient. To exclude mis-annotated TEs, the genes with mCHG in the WT (>0.05) are excluded from the analysis in panels **b**–**d** and **f** (excluded *n* = 1562; analyzed *n* = 26,723). Source data underlying Fig. 3c, d are provided as a Source Data file.

the recovery is much slower than the cases in which methylation remains in one of the two contexts[18,31,43]. Thus, each of mCH and mCG facilitates the establishment of the other.

Although such positive feedback would stabilize and enhance silent and active states, positive feedback alone would have a risk for the system to go out of control to excess. In addition to these local positive feedback mechanisms, our observation of *ibm1*, *met1*, and *ddm1* mutants revealed global negative feedback mechanisms to control genomic mCH levels. Most significantly, global loss of mCG induces strong enhancement of machinery to direct mCH to regions with mCG (Fig. 3; Supplementary Fig. 3). In addition, mCH is also controlled negatively by the global mCH level (Fig. 5). Global negative feedback, combined with local

positive feedback, would generate robust and balanced differentiation of active and silent genomic regions. The reaction-diffusion model is a powerful paradigm to understand pattern formation during development[44], but that could also be applied to the pattern formation of active and inactive transcription units within the genome. The positive feedback mechanisms function locally to separate heterochromatic and euchromatic genomic domains, and the negative feedback by a diffusible factor(s) controls the proportion of heterochromatic regions within the genome.

Differentiation between genes and TE genes depends on IBM1 as well as on a chromatin remodeler DDM1; DDM1 is necessary for the TE-specific mCG, mCH, and H3K9me[11,37,38,45], but the

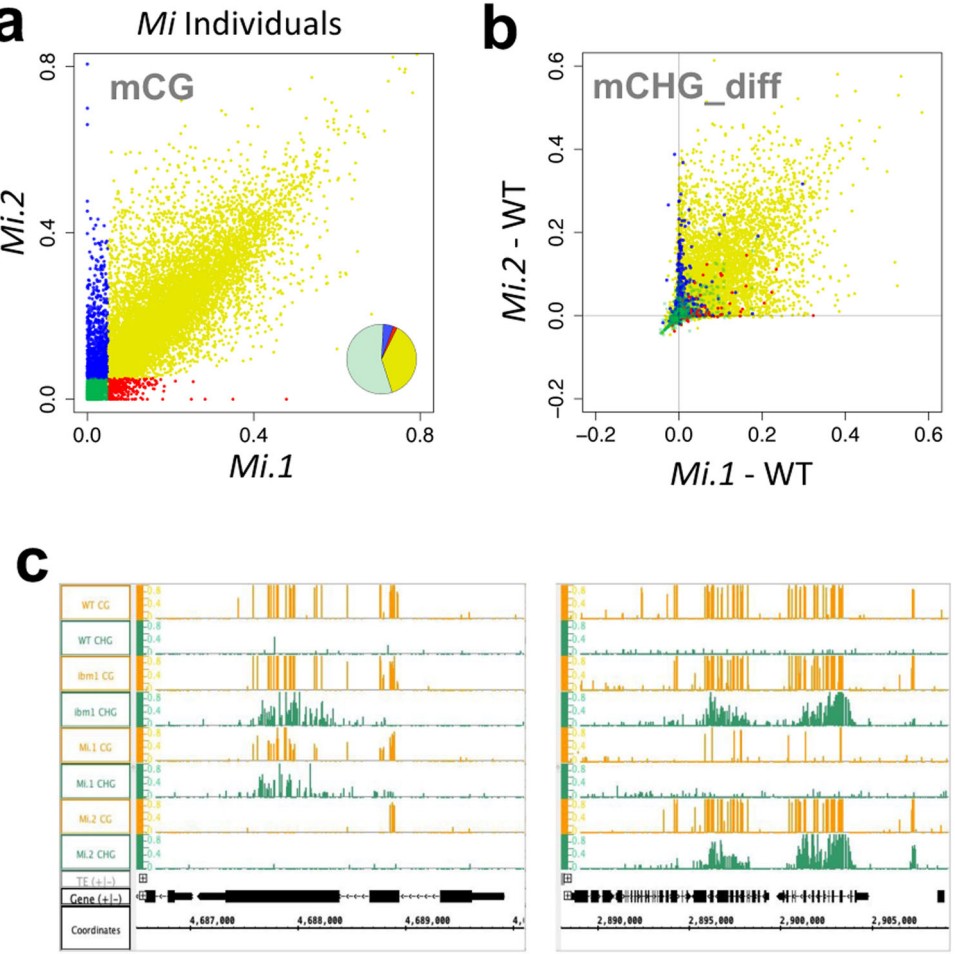

**Fig. 4 Association of genic mCG and mCH in epigenetic variant individuals. a** Differential mCG levels of genes in the two *Mi* individuals. The genes with the mCG presence (>0.05) or absence (<0.05) in the individuals are colored in yellow (commonly CG methylated), green (commonly CG hypomethylated), red (CG methylated only in *Mi*.1 individual), and blue (CG methylated only in *Mi*.2 individual). The ratio of each group was shown as pie chart at the right bottom. **b** The increase in genic mCHG in the *Mi* individuals. The genes are colored according to the groups in (**a**). **c** Browser view of representative variable ectopic mCHG in *Mi* individuals. The gene in the left panel shows ectopic mCHG in *ibm1* and *Mi*.1, but not in *Mi*.2. The genes in the right panel show genic mCHG in *ibm1* and *Mi*.2 but not in *Mi*.1. The regions of Chr5/4,686,500–4,690,100 (left) and Chr1/2,888,000–2,906,200 (right) are shown. Gray and black arrows below represent TEs and genes, respectively. To exclude the mis-annotated TEs, the genes with mCHG in the WT (>0.05) are excluded from the analysis in (**a**, **b**) (excluded $n = 1562$; analyzed $n = 26,723$). Source data underlying Fig. 4a, b are provided as a Source Data file.

underlying mechanisms have remained enigmatic. It has recently been shown that DDM1 binds to heterochromatin-specific H2A variant H2A.W and silences TE genes in combination with H2A.W[46]. This pathway seems to be conserved in mammals; LSH1, the mammalian ortholog of DDM1, silences repetitive sequences by deposition of heterochromatin-specific H2A variant macro-H2A[47]. In Arabidopsis, another H2A variant, H2A.Z, negatively interacts with mCG[18,41,48,49], and this pathway seems to be conserved to vertebrates[23]. The involvement of H2A variants in the local and global feedback mechanisms may also be an important target for future research.

In this work, we detected local and global crosstalk to reprogram DNA methylome. An important future challenge would be to understand the crosstalk in the context of development. DNA methylome can differ among cell types including those in stem cells and germ lines[50–54]. Interestingly, histone demethylase IBM1 is expressed in stem cells and reproductive cells[25]. In the future, dynamics would further be clarified by the single-cell genomics using preequilibrium materials, such as embryos of the first generation mutant of *IBM1*, *DDM1*, or other chromatin-modifying genes, as well as outcrossed progeny of these mutants.

## Methods

**Plant materials and growth conditions**. The mutants *met1-1*[28], *cc* (*cmt2 cmt3*)[11] and *sss* (*suvh4 suvh5 suvh6*)[55] are kind gifts from Eric Richards, Daniel Zilberman, and Judith Bender. The mutants *mcc*, *msss* (*met1-1 suvh4 suvh5 suvh6*), *m*F1, *mi* (*met1-1 ibm1-4*), *Mi* (MET1 *ibm1-4*), and *ddm1 ibm1* were made in this study. To create the mutants *mcc*, *msss*, and *m*F1 plants, *met1-1* and *cc* or *met1-1* and *sss* were genetically crossed, respectively. The F1 plants were self-pollinated, and the resulting F2 plants with all homozygous mutants of *met1-1 cmt2 cmt3* (*met1-1 cmt2 cmt3*) and *msss* (*met1-1 suvh4 suvh5 suvh6*) were selected and used for genetic crosses to create the *m*F1 plants. The plants were grown at 22 °C (16 h light, 8 h dark), firstly on MS agar media for 1–2 weeks, and then grown on soil. *ddm1 ibm1* plants were grown on MS agar media until harvesting.

**Whole-genome bisulfite sequencing and data processing**. Genomic DNA was extracted from rosette leaves of one individual plant using the Nucleon Phytopure genomic DNA extraction kit (GE Healthcare), and whole-genome bisulfite sequencing (WGBS) was performed[18]. Basically, two independent biological replicates were taken except for the wild-type Col-0, *met1-1*, and the second generation of *ibm1-4* mutant. Genomic DNA was subjected to fragmentation using a Focused Ultrasonicator (Covaris S220), and the size of 300–450 bp was gel-extracted. The libraries were prepared using TruSeq DNA LT Sample Prep Kit (Illumina) and then subjected to bisulfite conversion using MethylCode Bisulfite Conversion Kit (Life Technologies). The resulting DNA was amplified using KAPA HiFi HotStart Uracil ReadyMix (Kapa Biosystems) and purified with Agencourt AMPure XP (Beckman Coulter). Raw sequence data and processed data were

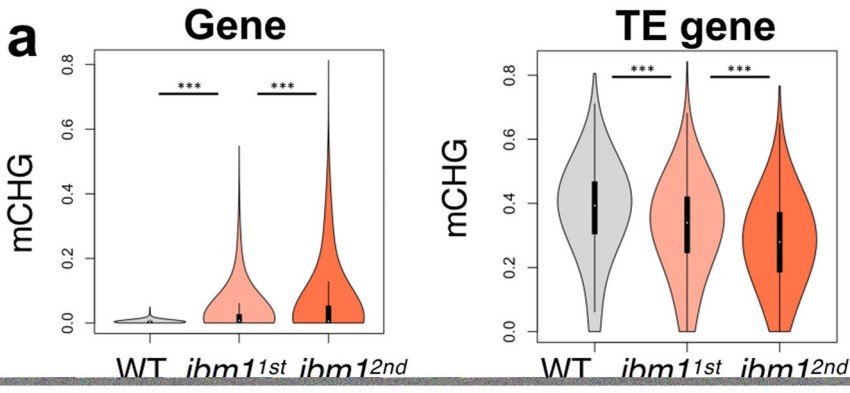

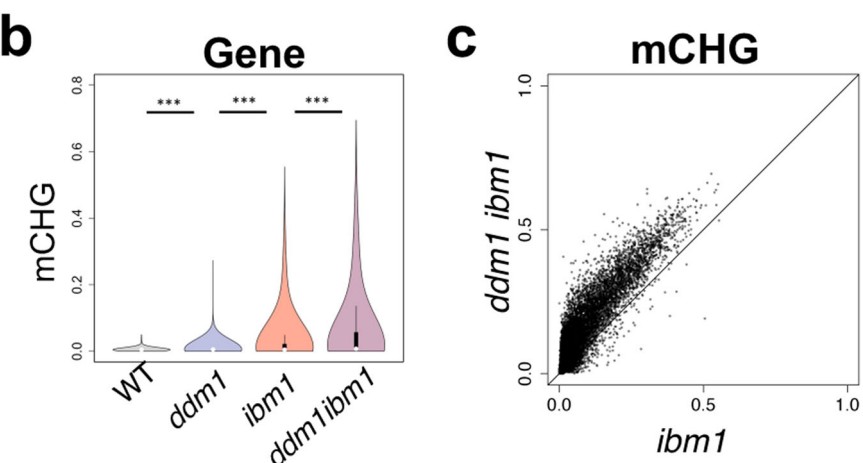

**Fig. 5 mCH is controlled not only locally but also globally. a** Violin plots showing the progressive accumulation of mCHG in genes and decrease of mCHG in TE genes in the *ibm1* mutant (***$P < 0.0001$, Wilcoxon signed-rank test, two-sided). **b** Violin plot showing that *ddm1* mutation further enhanced the *ibm1*-induced accumulation of mCHG in genes (***$P < 0.0001$, Wilcoxon signed-rank test, two-sided). **c** The *ibm1*-induced genic mCHG was globally enhanced in *ddm1 ibm1* double mutant (line: $y = x$). To exclude the mis-annotated TEs, the genes with mCHG in the WT (>0.05) are excluded from the analysis in (**a–c**) (excluded $n = 1562$; analyzed $n = 26,723$). Source data underlying Fig. 5 are provided as a Source Data file.

deposited in the GEO (GSE181896). The adaptor trimming and quality filtration were performed using Trimmomatic version 0.33[56]. The trimmed sequences were mapped to the Arabidopsis reference genome (TAIR10)[57], deduplicated, and methylation data extracted using Bismark version 0.10.157[58]. The annotations of genes and TEs are based on The Arabidopsis Information Resource[57]. The details of the annotation of TE genes are on the TAIR website (https://www.arabidopsis.org). We used Perl scripts[18] to count the numbers of methylated and total cytosines within a gene or within a 100-bp bin (For Supplementary Fig. 2a, b), and the methylation level for each context of each gene was calculated as the number of methylated cytosine within a gene divided by the number of total cytosines (weighted methylation level[59]). The numbers of methylated and unmethylated cytosines within a gene were used to extract the TE genes with statistically lower or higher mCH in *m*F1 than in F1 in Supplementary Fig. 3 by using edgeR (v3.28.1)[60] following the method in Chen et al.[61]. Two replicates of F1[18] and four replicates of *m*F1 were analyzed. The extracted TE genes were further filtered by their methylation difference between *m*F1 and F1 (>0.1 for mCHG, >0.03 for mCHH), and between WT and *met1-1* (<0.1 for mCHG, <0.03 for mCHH) to exclude the TE genes for which the significant methylation change is due to *met1-1* mutation. Rstudio (v1.1.463) was used to create scatter plots, box plots, and to perform statistical analysis. Genome Browser (Integrated Genome Browser[62]) was used for browser views. To create the heatmaps, TE genes and their surrounding regions (2 kb) were divided into 20 and 10 segments respectively, then for each segment, the value of methylated cytosines over total cytosines were calculated with Perl script. TE genes with the length (<1000, $n = 1038$) were excluded. The processed data were visualized using TreeView 3[63]. The methylation recovery rates shown in Fig. 1h, n, the methylation level of each gene in the F1 or *m*F1 were divided by that of corresponding background (WT or *met1-1*, respectively). To avoid division by values near zero, TE genes with low methylation level in WT and *met1-1* (CHG < 0.1 for Fig. 1h; $n = 318$, or CHH < 0.03 for Fig. 1n; $n = 388$) were excluded.

**RNA sequencing and data processing**. Plants were surface-sterilized, grown on MS agar medium, and harvested after 12 days. Total RNAs were purified using

TRIzol RNA Isolation Reagents (Thermo), and treated with DNase I (TaKaRa). The sequencing libraries were generated using the KAPA Stranded RNA-seq Library Preparation Kit, with RNA fragmentation for 7 min at 94 °C. The sequencing was performed in a 150-bp pair-end. The obtained data were quality filtered by using Trimmomatic (v0.33)[56], and the reads were mapped to genes through STAR algorithm[64], followed by bedtools[65]. After RPKM normalization, Rstudio (v1.1.463) was used to plot the expression levels. Two independent biological replicates were analyzed for each genotype.

**Reporting summary**. Further information on research design is available in the Nature Research Reporting Summary linked to this article.

## Data availability
WGBS and RNA-seq data generated in this study were deposited in the GEO with the accession number GSE181896. WGBS data for *cc*, *sss*, and F1 is available in the GEO with the accession number GSE148753. TAIR10 (https://www.arabidopsis.org) was used as the Arabidopsis reference genome. Source data are provided with this paper.

## Code availability
Perl script for DNA methylation counting per gene can be available from GitHub.

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

## Acknowledgements

We thank Eric Richards and Robert Schmitz for critical comments on the manuscript, and Judith Bender, Robert Fischer, Eric Richards, and Daniel Zilberman for sharing mutant strains. Computations were partially performed on the NIG supercomputer at NIG, Japan. Supported by grants from, Japanese Ministry of Education, Culture, Sports, Science and Technology (26221105, 15H05963, 19H00995, and 21H04977 to T.K., 19H05740 and 17K15059 to T.K.T.), CREST Grant, Japan (JPMJCR15O1 to T.K.), Systems Functional Genetics Project of the Transdisciplinary Research Integration Center, ROIS, Japan (to Y.T. and T.K.).

## Author contributions

T.K.T., C.Y., S.O., and T.K. designed the study. T.K.T., C.Y., S.O., S.T., A.K., Y.T., and T.K. performed the experiments. T.K.T., C.Y., and S.O. analyzed the data. T.K.T. and T.K. wrote the paper incorporating comments from the other authors.

## Competing interests

The authors declare no competing interests.
