## [Peer Review File · Nature Communications]

Reviewers' Comments:

Reviewer #1:

Remarks to the Author:

This manuscript follows up on a previous study by the same group, which showed that non-CG methylation and H3K9me2 are efficiently restored at TE coding sequences after these marks are lost (To et al Nat Plants 2020). This study suggested a link between CG methylation and the efficiency of CH methylation recovery.

The current manuscript directly assesses this link. Using the hypomorphic *met1-1* mutant allele, which does not induce complete loss of CG methylation throughout the genome, the authors show that CH methylation can only be restored at TE genes with residual CG methylation in *met1*, whereas total loss of CG methylation abolishes non-CG methylation recovery. The authors further show that the relative level of CG methylation is important for the targeting of CH methylation at genes in the absence of IBM1 function (which targets protein-coding genes). Gain of CH methylation in *ibm1* mutants is associated with reduced levels of non-CG methylation at TE genes. The authors also show that gain in CH methylation at genes is much enhanced when both DDM1 (which targets heterochromatic TEs) and IBM1 are lacking. Based on these and previous observations, the authors propose a model wherein epigenome patterning relies on both local positive and global negative feedbacks.

These findings provide important new insights into the interaction between CG and non-CG methylation. I really enjoyed reading this manuscript. It is clearly written, the genetic approaches are simple and clever and the data sound.

I have only minor comments/suggestions:

- The authors make a big use of scatter plots to illustrate pairwise comparisons of weighted methylation levels between genotypes. Given the huge number of points in each plot, it would be helpful to the reader, not only to color, but also to include the numbers of loci showing statistically significant variation between the considered genotypes.
- The global negative feedback model presented in Fig. 6b is rather complex. It would be nice to find an even more visual/schematic way to illustrate it.
- The analyses in the current paper are restricted to DNA methylation. Therefore, I would suggest the authors refer to "methylome" instead of "epigenome" in the title and abstract.
- In Fig. 2a, the labels for F1s are inconsistent (mF1 vs. F1(*met1*))

Reviewer #2:

Remarks to the Author:

This manuscript uncovers some of the mechanisms implicated in heterochromatin dynamics in plants. The manuscript is relatively well written, but it may benefit with some grammatical corrections.

Overall, the story is logically presented and the conclusions reached are well supported with data. I only have minor suggestions that may improve the quality of the manuscript.

(1) Some references (eg. 18) are repeated throughout the text. Also review allele names to ensure they are correct.

(2) Authors state in several occasions "relative, rather than absolute, level of mCG..." but this description should be clarified.

(3) Provide statistical significance values and details of test used in Fig 1H and Fig 1N.

(4) Have TE length been normalized by length in Fig 2B. What is the flanking window used in this plot? What is the methylation value represented here, are values absolute/ Log?

(4) Label axis in line plots and statistical values for boxplots in Fig 3B and C respectively.

(5) Provide statistical significance values and details of test used in violin plots of Fig 5.

(6) Is the model in Figure 6 necessary? Most of the points are already covered in the discussion.

I wonder if the authors have considered investigating the phenotypic consequences of this

local/positive and global/negative feedback proposed. Perhaps a transcriptome analysis, with some of the genetic materials generated, could shed light on the biological significance of these processed in coding/non-coding features of the genome.

Reviewer #3:

Remarks to the Author:

The manuscript "Local and global crosstalk among heterochromatin marks drives epigenome patterning in Arabidopsis" by To and co-workers describes the re-establishment of DNA methylation after re-introduction of genetically lost DNA methylation pathways. To do so, the authors use an elegant genetic approach, taking advantage of the interdependency of histone H3 lysine 9 dimethylation (H3K9me₂) and DNA methylation in the CHG and CHH contexts (mCH). Hence, a triple mutant in the histone methyltransferases (MTases) SUVH4/5/6 leads to loss of mCH in the same extent as mutants in the CHH and CHG maintenance MTases CMT2 and CMT3. The authors have previously shown that non-CG methylation is readily re-established using this system and this process is largely independent of the RNA directed DNA methylation (RdDM) pathway in coding regions of transposable elements (TEs) and is found mainly over regions that show methylation in the CG context (To et al, Nature Plants 2020). In addition, and in line with another previous publication by the Kakutani group (Ito et al, PLoS Genetics 2015), the authors report that loss of mCH results in ectopic mCH at novel sites, suggesting that negative feedback regulation is acting on a genome-wide level. This is an interesting contribution to our understanding of DNA methylation targeting, suggesting that mCG guides mCH, and thereby providing a positive feedback-loop. Yet, this has been previously published by the Kakutani group as well as others (To et al, Nature Plants 2020; Catani et al, EMBO J 2017). The novelty presented in this manuscript is that the re-establishment of mCH follows relative rather than absolute mCG levels, although it remains unclear how this is mechanistically achieved. Previous work has referred to histone H2A variants that discriminate genes from TEs, but this has not been further exploited in this report. On the other side, the negative feedback control to maintain global DNA methylation levels infers (epi)genome surveillance mechanisms that are largely elusive. The results are well presented, the figures clear and the manuscript is well written. Except for some minor points (see below), the methods seem sound and are appropriately described. Points requiring further investigation are listed in the following:

- 1) Given that "Transposable elements (TEs) are robustly silenced by targeting of multiple epigenetic marks", as stated by the authors, it would be desirable to show transcriptional analysis of TEs in the lines established in this publication. To which extent is the remaining CG methylation level in met1-1 in combination with the remethylation of CH sites sufficient to repress the associated TEs? This information might also reveal if the absolute or the relative mCG is important for its repressive function(see also below).
- 2) Establishment of met1-1 cmt2/3 (mcc) and met1-1 suvh4/5/6 (msss) mutants -> There is no information concerning how these mutants have been established. Does it make a difference if met1-1 and cmt2/3 have been crossed as homo- or heterozygous parental lines and if so, in which sense does this impact the results? Previous work by the Kakutani group has shown that it matters which gene knock-out is established first in combination of different histone h2a alleles with cmt2/3 mutants (To et al, Nature Plants 2020). For instance, it is known that met1 and ddm1 mutants confer stable DNA hypomethylation in backcrosses to wild type. This would mean that a met1 mutant used for crossing would be epigenetically different from the wild type.
- 3) The observation that relative rather than absolute mCG dictates mCH is exciting and raises conceptual issues, given that a single cell is expected to have a rather digital mCG pattern for a specific CG position (methylation on either copies, one copy or none of the copies in a diploid state = full methylation, half methylation or no methylation). In fact, CG methylation in flowering plants and mammals seems to exist in either full methylated or non-methylated state in a given tissue. First of

all, the authors compare the methylation level of the entire annotation. How does the size of the annotation relate to the DNA methylation level/remethylation? For instance, TE genes are globally larger than other TE sequences, which seems to coincide with the CH remethylation capacity. How does the correlation look like when the authors use segments of a fixed size (100-200 bp for example) instead of entire annotations? Second, the methylomes have been generated from leaves that contain many different cell types and are derived from the meristem, which is known to show elevated RdDM and TE silencing activity (Baubec et al, EMBO Rep 2014). Therefore, it might be conceivable that mCG and mCH initiation takes place in the meristems due to RdDM and that the maintenance fails outside the stem cells due to the met1-1 mutation. In this way, the proposed relative distribution of mCG (and as a consequence mCH) might depend on the tissue used, e.g. low levels in leaves of met1-1 but possibly high levels in the meristems due to elevated RdDM activity there. This should be taken into account.

4) It is not entirely clear to this reviewer how the increase in CHG methylation in consecutive generations of *ibm1/ibm1 ddm1* mutants presented in last paragraph relates to the rest of the story. The negative feedback model has been presented previously and I am not sure to which extent the *ddm1 ibm1* really reveals something new here. E.g. how does this relate to the CH remethylation guided by mCG in the previous parts? This should be exploited more thoroughly. For instance, to which extent does the increase in mCHG in *ibm1* or *ibm1 ddm1* overlap with de novo mCHG observed in *mF1* or *miF1*? Moreover, mCHH has been explored in combination with mCHG in the previous section, but here the authors only present data on mCHG. Therefore, how does mCHH behave in this background? Finally, why did the authors use further generations of *ibm1* but not *met1-1*? The authors mention that the hypomethylation increases in consecutive generations in *met1* and hence, one might expect a similar establishment of ectopic mCH in this scenario as was described for *ibm1 / ddm1* with the advantage to make it more comparable with the other parts. In this way, the relationship between regions that lose and those that gain mCH/mCG in the presented genetic backgrounds might reveal some hints towards a more mechanistic interpretation.

Point-by-point response to Reviewers comments

(Reviewers comments and our responses are shown in black and blue, respectively)

Reviewer #1 (Remarks to the Author):

Reviewer #1 is very positive. Nonetheless, incorporation of constructive suggestions of this reviewer has strengthened the manuscript very much.

This manuscript follows up on a previous study by the same group, which showed that non-CG methylation and H3K9me2 are efficiently restored at TE coding sequences after these marks are lost (To et al Nat Plants 2020). This study suggested a link between CG methylation and the efficiency of CH methylation recovery.

The current manuscript directly assesses this link. Using the hypomorphic *met1-1* mutant allele, which does not induce complete loss of CG methylation throughout the genome, the authors show that CH methylation can only be restored at TE genes with residual CG methylation in *met1*, whereas total loss of CG methylation abolishes non-CG methylation recovery. The authors further show that the relative level of CG methylation is important for the targeting of CH methylation at genes in the absence of IBM1 function (which targets protein-coding genes). Gain of CH methylation in *ibm1* mutants is associated with reduced levels of non-CG methylation at TE genes. The authors also show that gain in CH methylation at genes is much enhanced when both DDM1 (which targets heterochromatic TEs) and IBM1 are lacking. Based on these and previous observations, the authors propose a model wherein epigenome patterning relies on both local positive and global negative feedbacks.

These findings provide important new insights into the interaction between CG and non-CG methylation. I really enjoyed reading this manuscript. It is clearly written, the genetic approaches are simple and clever and the data sound.

Thank you for the very positive evaluation.

I have only minor comments/suggestions:

- The authors make a big use of scatter plots to illustrate pairwise comparisons of weighted methylation levels between genotypes. Given the huge number of points in each plot, it would be helpful to the reader, not only to color, but also to include the

numbers of loci showing statistically significant variation between the considered genotypes.

As suggested, we extracted the TE genes with statistically lower mCH in *mF1* than in F1 using edgeR (Robinson et al., 2010; Chen et al., 2017). We colored them red in the scatter plot and put the number of TE genes in Supplementary Fig. 3a,b. In addition to the scatter plot, we show their properties by box plots for mCH levels among considered genotypes (Supplementary Fig. 3c–f). Details of the analyses including statistics are added to the Methods section.

During these analyses, we realized interesting properties of TE genes with stronger mCG-directed mCH recovery in *mF1* than in F1 (blue dots in Supplementary Fig. 3a,b); in those TEs, mCG level tends to be lower in *mF1* than in F1 (Supplementary Fig. 3e,f, ‘mCG’ panels), but relative mCG within the *mF1* genome is high, but not in the F1 (Supplementary Fig. 3e,f, ‘RANK’ panels). Thus, relative mCG level within the genome seems to be important for the mCH recovery in TE genes; this conclusion in TE genes is consistent with that on genic mCH induced by *ibm1*. These analyses strengthened our proposal that mCG-directed mCH establishment is controlled by global negative feedback; that can be detected not only for ectopic genic mCH in *ibm1*, but also for mCH in TE genes. We thank the reviewer for the suggestion.

- The global negative feedback model presented in Fig. 6b is rather complex. It would be nice to find an even more visual/schematic way to illustrate it.

We agree that Fig 6b was rather complex. In addition, as the Reviewer #2 suggested (#6), most of the points in Fig 6 are already covered in the discussion. We therefore removed this Figure, as the Reviewer #2 suggested.

- The analyses in the current paper are restricted to DNA methylation. Therefore, I would suggest the authors refer to “methyloome” instead of “epigenome” in the title and abstract.

As suggested, we changed the “epigenome” in the title and abstract to “DNA methyloome”.

- In Fig. 2a, the labels for F1s are inconsistent (mF1 vs. F1(met1))

We corrected the inconsistency. We thank the reviewer for pointing that out.

Reviewer #2 (Remarks to the Author):

Reviewer #2 is also very positive. We thank the reviewer for the constructive suggestions, because incorporation of these suggestions has significantly strengthened the manuscript and made it easier to follow.

This manuscript uncovers some of the mechanisms implicated in heterochromatin dynamics in plants. The manuscript is relatively well written, but it may benefit with some grammatical corrections. Overall, the story is logically presented and the conclusions reached are well supported with data. I only have minor suggestions that may improve the quality of the manuscript.

Thank you for the very positive evaluation.

(1) Some references (eg. 18) are repeated throughout the text. Also review allele names to ensure they are correct.

We confirmed allele names and references carefully. We corrected the typo of allele name “*met1-*” to “*met1-1*” in the Results section. We also reorganized the manuscript to make the parts describing To et al (2020) more compact and easy to follow.

(2) Authors state in several occasions "relative, rather than absolute, level of mCG..." but this description should be clarified.

In order to clarify the meaning, we modified the expression “relative, rather than absolute, level of mCG” to the following:

“relative, rather than absolute, level of mCG within the genome” (within Abstract)

“relative level of mCG within the genome, rather than the absolute levels of mCG” (last paragraph in Introduction)

“relative, rather than absolute, level of mCG within the genome is critical. In other words, mCH is targeted to regions with mCG, but this targeting seems much enhanced when mCG levels are low in the other regions of the genome.” (within Results)

(3) Provide statistical significance values and details of test used in Fig 1H and Fig 1N.

We performed statistical analysis (Wilcoxon signed rank test) and added that information in Fig 1.

(4) Have TE length been normalized by length in Fig 2B. What is the flanking window used in this plot? What is the methylation value represented here, are values absolute/Log?

Thank you for the confirmation and suggestion. In Fig 2b, TE gene length had been normalized by the length. The flanking region is 2kb, and the methylation values are absolute (from 0 to 1). We added these information to Fig. 2b.

(4) Label axis in line plots and statistical values for boxplots in Fig 3B and C respectively.

As suggested, we added the axis and statistical values.

(5) Provide statistical significance values and details of test used in violin plots of Fig 5.

As suggested, we performed the statistical analysis and added that in Fig 5.

(6) Is the model in Figure 6 necessary? Most of the points are already covered in the discussion.

We removed the Figure 6, because we agree with the reviewer that most of the points are already covered in the discussion.

I wonder if the authors have considered investigating the phenotypic consequences of this local/positive and global/negative feedback proposed. Perhaps a transcriptome analysis, with some of the genetic materials generated, could shed light on the biological significance of these processed in coding/non-coding features of the genome.

As suggested, we performed transcriptome analysis of several genotypes used in Fig. 1 and Fig. 2. Simultaneous loss of mCG and mCH in *msss* or *mcc* mutants results in transcriptional derepression in many TE genes. Importantly, the mCG-dependent mCH

recovery in the *mF1* (Fig. 2d) induces robust re-silencing of them (Fig. 2e), demonstrating the biological importance of mCH recovery in TE silencing.

Reviewer #3 (Remarks to the Author):

Reviewer #3 is also very positive overall. We tried to incorporate suggestions of the Reviewer #3 as much as possible, which improved the manuscript significantly. We thank the Reviewer.

The manuscript “Local and global crosstalk among heterochromatin marks drives epigenome patterning in Arabidopsis” by To and co-workers describes the re-establishment of DNA methylation after re-introduction of genetically lost DNA methylation pathways. To do so, the authors use an elegant genetic approach, taking advantage of the interdependency of histone H3 lysine 9 dimethylation (H3K9me2) and DNA methylation in the CHG and CHH contexts (mCH). Hence, a triple mutant in the histone methyltransferases (MTases) SUVH4/5/6 leads to loss of mCH in the same extent as mutants in the CHH and CHG maintenance MTases CMT2 and CMT3. The authors have previously shown that non-CG methylation is readily re-established using this system and this process is largely independent of the RNA directed DNA methylation (RdDM) pathway in coding regions of transposable elements (TEs) and is found mainly over regions that show methylation in the CG context (To et al, Nature Plants 2020). In addition, and in line with another previous publication by the Kakutani group (Ito et al, PLoS Genetics 2015), the authors report that loss of mCH results in ectopic mCH at novel sites, suggesting that negative feedback regulation is acting on a genome-wide level.

This is an interesting contribution to our understanding of DNA methylation targeting, suggesting that mCG guides mCH, and thereby providing a positive feedback-loop.

We thank the reviewer for appreciating the importance of the conclusion that mCG guides mCH.

Yet, this has been previously published by the Kakutani group as well as others (To et al, Nature Plants 2020; Catani et al, EMBO J 2017).

As the reviewer pointed out, we have previously shown correlation between mCG and ability to target mCH (To et al 2020). Importantly, however, causative link remained unexplored. In current manuscript, we directly tested and proved the causative link. We also showed the causative link for the mCG-guided genic mCH seen in the *ibm1* background. Catoni et al examined recovery of mCG, not mCH; so, they examined reverse pathway, which depends on RNAi and much less efficient than the pathways described in this paper. In order to clearly mention that causative link has not been tested previously, we added the phrase “although the causative link between them remains to be tested” at the end of the third paragraph of Introduction.

The novelty presented in this manuscript is that the re-establishment of mCH follows relative rather than absolute mCG levels, although it remains unclear how this is mechanistically achieved.

Thank you for appreciating novelty of the negative feedback. We additionally show very similar effect for the mCG-directed mCH in TE genes (Fig. 2d and Supplementary Fig. 3e,f). That has strengthened the manuscript very much, as stated more explicitly below (in our response third from the end).

Previous work has referred to histone H2A variants that discriminate genes from TEs, but this has not been further exploited in this report. On the other side, the negative feedback control to maintain global DNA methylation levels infers (epi)genome surveillance mechanisms that are largely elusive.

As the Reviewer pointed out, we previously reported that ability to recover mCH is associated with H2AW as well as mCG; TE genes with replacement of H2AW to H2AZ did not show efficient mCH recovery (To et al 2020). In our preliminary results, however, although presence of H2AW correlates with the mCH recovery, mutation of H2AW has shown only limited effect on the ability to recover mCH; association does not necessarily mean causative link. However, given the preliminary nature of these observations, we would like to refrain from including these preliminary results in this paper. We would rather like to make a more complete story later, after carefully characterizing possible impacts of this and other H2A variants by multiple additional approaches.

The results are well presented, the figures clear and the manuscript is well written.

Except for some minor points (see below), the methods seem sound and are appropriately described.

We thank the Reviewer #3 for this very positive evaluation.

Points requiring further investigation are listed in the following:

Given that “Transposable elements (TEs) are robustly silenced by targeting of multiple epigenetic marks”, as stated by the authors, it would be desirable to show transcriptional analysis of TEs in the lines established in this publication. To which extent is the remaining CG methylation level in *met1-1* in combination with the remethylation of CH sites sufficient to repress the associated TEs? This information might also reveal if the absolute or the relative mCG is important for its repressive function (see also below).

Thank you very much for the constructive suggestion. As suggested, we performed transcriptome analysis of several genotypes used in Fig. 1 and Fig. 2. Simultaneous loss of mCG and mCH in *msss* or *mcc* mutants results in transcriptional derepression in many TE genes. Importantly, the mCG-dependent mCH recovery in the *mF1* (Fig. 2d) induces robust re-silencing of them (Fig. 2e), demonstrating the biological importance of mCH recovery in TE silencing.

2) Establishment of *met1-1 cmt2/3* (*mcc*) and *met1-1 suvh4/5/6* (*msss*) mutants →
There is no information concerning how these mutants have been established. Does it make a difference if *met1-1* and *cmt2/3* have been crossed as homo- or heterozygous parental lines and if so, in which sense does this impact the results? Previous work by the Kakutani group has shown that it matters which gene knock-out is established first in combination of different histone h2a alleles with *cmt2/3* mutants (To et al, Nature Plants 2020). For instance, it is known that *met1* and *ddm1* mutants confer stable DNA hypomethylation in backcrosses to wild type. This would mean that a *met1* mutant used for crossing would be epigenetically different from the wild type.

As suggested, we added description of the order of fixation of the mutations are added to the Methods section. In the system of Fig 6 in To et al 2020, we examined effects of combining *h2az* and *cmt* mutations, which affect mCG in opposite ways. Due to the opposite effects, we suspected that order of fixation can be important. In *msss* and *mcc* mutant used in this manuscript, we could not find any specific reason to suspect that the

order of fixation affects their epigenome. Nonetheless, we agree the reviewer that it is desirable to add the information of the order of fixation in the Methods section.

3) The observation that relative rather than absolute mCG dictates mCH is exciting and raises conceptual issues, given that a single cell is expected to have a rather digital mCG pattern for a specific CG position (methylation on either copies, one copy or none of the copies in a diploid state = full methylation, half methylation or no methylation). In fact, CG methylation in flowering plants and mammals seems to exist in either full methylated or non-methylated state in a given tissue. First of all, the authors compare the methylation level of the entire annotation. How does the size of the annotation relate to the DNA methylation level/remethylation? For instance, TE genes are globally larger than other TE sequences, which seems to coincide with the CH remethylation capacity. How does the correlation look like when the authors use segments of a fixed size (100-200 bp for example) instead of entire annotations?

As suggested, we compared TE length and mCH levels in F1 and *m*F1. Similar effects were seen on mCH, irrespective of the length. That was added to Supplementary Fig. 2c. We also analyzed them in the segments of fixed size (100bp) as suggested. Clear correlation between the mCG level and mCH recovery was seen as is the case in the analyses done in transcription units. The results are added to Supplementary Fig. 2a,b.

Second, the methylomes have been generated from leaves that contain many different cell types and are derived from the meristem, which is known to show elevated RdDM and TE silencing activity (Baubec et al, EMBO Rep 2014). Therefore, it might be conceivable that mCG and mCH initiation takes place in the meristems due to RdDM and that the maintenance fails outside the stem cells due to the *met1-1* mutation. In this way, the proposed relative distribution of mCG (and as a consequence mCH) might depend on the tissue used, e.g. low levels in leaves of *met1-1* but possibly high levels in the meristems due to elevated RdDM activity there. This should be taken into account.

As the reviewer pointed out, the difference of methylation machineries among cell types is an important issue. We added a new paragraph at the end of Discussion section, to discuss the reprogramming in the context of development. Resolving these questions will need further investigation in future. We thank the reviewer, because that discussion has strengthened the manuscript.

In regard to the possibility that RdDM in meristem plays part in reprogramming, our previous study showed that the CH recovery in TE genes is largely independent of RdDM pathway; TE genes recover mCH in *drm2* or *rdr1 rdr2 rdr6* triple mutant backgrounds (To et al., 2020).

4) It is not entirely clear to this reviewer how the increase in CHG methylation in consecutive generations of *ibm1/ibm1 ddm1* mutants presented in last paragraph relates to the rest of the story.

The negative feedback model has been presented previously and I am not sure to which extent the *ddm1 ibm1* really reveals something new here. E.g. how does this relate to the CH remethylation guided by mCG in the previous parts? This should be exploited more thoroughly.

As suggested, we added explanation for the relationship of the *ddm1 ibm1* results to the rest of the story in the last paragraph of Results section. The *ddm1*-induced loss of mCH in TE genes induces enhancement of the *ibm1*-induced genic mCH (Figure 5bc). The results are complementary to the results for the negative effect of the *ibm1*-induced genic mCH to TE mCH (Figure 5a). By this further explanation, the story become clearer. We thank the reviewer for the suggestion.

For instance, to which extent does the increase in mCHG in *ibm1* or *ibm1 ddm1* overlap with de novo mCHG observed in *mF1* or *miF1*?

As we discussed in our response above, we examined the negative feedback in the *mF1*. Interestingly, mCH recovery is higher in *mF1* than in *F1* in some of the TE genes (blue dots in Supplementary Fig. 3a,b); in those TEs, mCG level tends to be lower in *mF1* than in *F1* (Supplementary Fig. 3e,f, 'mCG' panels), but relative mCG within the *mF1* genome is high, but not in the *F1* (Supplementary Fig. 3e,f, 'RANK' panels). Thus, relative mCG level within the genome seems to be important for the mCH recovery in TE genes. This conclusion is consistent with that on genic mCH induced by *ibm1*. These analyses strengthened our proposal that mCG-directed mCH establishment is controlled by global negative feedback; that can be detected not only for ectopic genic mCH in *ibm1*, but also for mCH in TE genes.

Moreover, mCHH has been explored in combination with mCHG in the previous section, but here the authors only present data on mCHG. Therefore, how does mCHH behave in this background?

As suggested, we added results for mCHH as Supplementary Fig 7.

Finally, why did the authors use further generations of *ibm1* but not *met1-1*? The authors mention that the hypomethylation increases in consecutive generations in *met1* and hence, one might expect a similar establishment of ectopic mCH in this scenario as was described for *ibm1 / ddm1* with the advantage to make it more comparable with the other parts. In this way, the relationship between regions that lose and those that gain mCH/mCG in the presented genetic backgrounds might reveal some hints towards a more mechanistic interpretation.

As the reviewer mentioned, genic mCHG increases progressively in *ibm1* mutant, and that is the reason we showed results for the first and second generation of *ibm1*. We do not know if hypomethylation increases in consecutive generations in *met1*; we have not mentioned that in either this or previous manuscripts. We are now characterizing epigenetic inbred lines (epiRILs) from *met1*, which was generated in Paszkowski lab, in order to understand parameters affecting the establishment of mCG. But that is out of the scope of this manuscript.

Reviewers' Comments:

Reviewer #1:

Remarks to the Author:

The authors have performed additional experiments and appropriate and careful revisions that fully address my previous comments. I have no further comments on the manuscript, which I consider an excellent piece of work.

Reviewer #2:

None

Reviewer #3:

Remarks to the Author:

The revised version of the manuscript has improved and my remarks have been properly answered. The re-examination of the data for the recovery of mCH in the mF1 versus F1 background revealed some interesting novel insights. Remarkably, for regions that do not regain mCH in the mF1, the mCG level is lower in the mF1 compared to met1. Yet, regions that show elevated and low mCH, respectively in the mF1, show comparable mCG levels in met1. This, in turn suggests that it is not the initial mCG level in the met1 mutant that drives mCH, but that there is a re-establishment of mCG that relates to the mCH level in the mF1. This needs to be taken into account in the conclusions.

Apart from these general remarks there are only two minor points that might be worth revisiting:

1) „mCG also guides mCH in active genes, although genic mCH/H3K9me is removed.“ This sounds contradictory and should be rephrased.

2) “The spectrum of mCH and mCG [] differed even between different mi individuals.” This is an important point as it suggests that there is some arbitrary component involved in the establishment of proper DNA methylation. In this respect, it might be worth mentioning that the loss of mCG is not only affecting different sites, but is quantitatively different between the two Mi lines (Mi1 showing stronger mCG loss compared to Mi2).

Response to the comments by Reviewers

Reviewer #1 (Remarks to the Author):

The authors have performed additional experiments and appropriate and careful revisions that fully address my previous comments. I have no further comments on the manuscript, which I consider an excellent piece of work.

[Editor: Reviewer #2 states in Remark to Editor section that (s)he is satisfied with the revision.

We are very happy to hear that Reviewers #1 and #2 are satisfied with the revision.

Reviewer #3 (Remarks to the Author):

The revised version of the manuscript has improved and my remarks have been properly answered.

We are very happy to hear that.

The re-examination of the data for the recovery of mCH in the mF1 versus F1 background revealed some interesting novel insights. Remarkably, for regions that do not regain mCH in the mF1, the mCG level is lower in the mF1 compared to met1. Yet, regions that show elevated and low mCH, respectively in the mF1, show comparable mCG levels in met1. This, in turn suggests that it is not the initial mCG level in the met1 mutant that drives mCH, but that there is a re-establishment of mCG that relates to the mCH level in the mF1. This needs to be taken into account in the conclusions.

As the Reviewer #3 pointed out, in the TE genes with inefficient mCH recovery in mF1, mCG is generally lower in mF1 than in *met1-1* (Supplementary Fig 3c,d, center). Actually, the parental lines of the mF1 (i.e. *mcc* and *msss*) also show lower mCG than in *met1-1* (data not shown). The difference is likely to reflect dependence of mCG maintenance on mCH. As we agree with the Reviewer that this point is related to the conclusions, we added the following sentence in the Figure legend.

“In addition, in the TE genes with inefficient mCH recovery in *mF1*, mCG is lower in *mF1* than in *met1-1* (Supplementary Fig 3c,d, center), likely reflecting the role of mCH for the maintenance of mCG for these TE genes in the *met1-1* mutant background.”

1) „mCG also guides mCH in active genes, although genic mCH/H3K9me is removed.”
This sounds contradictory and should be rephrased.

Thank you for the comment. We rephrased it to “mCG also guides mCH in active genes, although the resulting mCH/H3K9me is removed thereafter.”

2) “The spectrum of mCH and mCG [] differed even between different *mi* individuals.”
This is an important point as it suggests that there is some arbitrary component involved in the establishment of proper DNA methylation. In this respect, it might be worth mentioning that the loss of mCG is not only affecting different sites, but is quantitatively different between the two *Mi* lines (*Mi1* showing stronger mCG loss compared to *Mi2*).

Thank you for the comment. As the Reviewer #3 pointed out, the loss of mCG in *Mi.1* is stronger than that in *Mi.2*. Thus, we modified the sentence “the spectrum of mCG differed among individual *Mi* plants” to “the spectrum and degree of mCG differed among individual *Mi* plants”.